# An Instagram Hashtag Fostering Science Education of Vulnerable Groups during the Pandemic

**DOI:** 10.3390/ijerph19041974

**Published:** 2022-02-10

**Authors:** Lídia Puigvert, Beatriz Villarejo-Carballido, Regina Gairal-Casadó, Aitor Gómez, Paula Cañaveras, Teresa Sordé Martí

**Affiliations:** 1Department of Sociology, Faculty of Economics and Business, University of Barcelona, 08034 Barcelona, Spain; paula.canaveras@ub.edu; 2Department of Theory and History of Education, University of Barcelona, 08007 Barcelona, Spain; beatrizvillarejo@ub.edu; 3Department of Pedagogy, Faculty of Education Sciences and Psychology, Rovira i Virgili University, 43007 Tarragona, Spain; regina.gairal@urv.cat (R.G.-C.); aitor.gomez@urv.cat (A.G.); 4Department of Sociology, Faculty of Political Science and Sociology, Autonomous University of Barcelona, 08193 Cerdanyola del Vallès, Spain; teresa.sorde@uab.cat

**Keywords:** Instagram, science education, vulnerable groups, pandemic, young people

## Abstract

Scientific literature presents young people as a vulnerable group at risk of poverty and social exclusion. One of the elements that have the most significant impact on reducing their vulnerability is promoting education. Little is known about how social networks can promote the education of young people. To address this, the present study aims to analyse how social networks, specifically Instagram, which is one of the most used by young people, has promoted, among other aspects, the scientific education of young people during the COVID-19 pandemic. This study analyses 5000 education-related Instagram posts made during the COVID-19 pandemic (March 2021) European research project ALLINTERACT. We have analysed those posts that show, on the one hand, how citizens benefit from scientific research and, on the other hand, citizens’ awareness of the impact of scientific research. Through the analysis of the posts, it has been observed how Instagram has been a social network that has provided information and scientific advances in various branches of knowledge, created knowledge networks, and provided a channel for information about the pandemic. Through the analysis of the 5000 posts, it is evident how Instagram has provided spaces for scientific learning, fostering access to scientific education for young people.

## 1. Introduction

One-third of young people in Europe are at risk of poverty and social exclusion [1]. The United Nations have also identified their high vulnerability as people who do not have equal access to social values and institutions [2]. In 1998, the same institution in the Declaration on Youth Policies and Programs in Lisbon [3] already presented a great international concern about the situation of young people, due to the numerous elements of vulnerability that surrounded them and that we have not yet solved. The Organisation for Economic Co-operation and Development (OECD) stated in 2015 that, in most partner countries, young people are more at risk of poverty and in 2016 highlighted that 40 million young people in OECD countries were at risk of not having employment, education, or training [4].

Studies show that, far from reducing the vulnerability of this group, the situation of risk in which they find themselves continues to increase, as the proportion of young people living in households with very low work intensity, which leads to situations of serious risk to their economic and social well-being, continues to rise [5].

Risk status is defined as an interactive process between the social contexts in which a person finds themselves and the set of risk factors that, when present, place young people at risk of social exclusion [6]. Contexts of vulnerability are highlighted as those in which young people had encountered or were in a situation of family disruption, domestic violence, geographical dislocation, unemployment, involvement in violence, susceptibility to early parenthood, sexually transmitted infections, and disease [2,3,4,5,6,7]. Other research defines the risk of living in vulnerability as related to demographic characteristics, interpersonal relationships, access to resources, individual capacity, and the availability of support, with the consideration of race and poverty as central factors [8,9,10]. Situations of vulnerability are exacerbated when young people do not have an individual, social and environmental context to cope with risky situations, lacking the skills, knowledge, or support to overcome them [11].

### 1.1. Promoting Education to Reduce Vulnerability

The scientific literature highlights that having a higher education reduces vulnerability and provides access to greater economic and social well-being, taking into account that the academic level required to achieve better living conditions is increasingly higher, with post-compulsory secondary education and university education becoming increasingly important [12,13,14]. Data show that at the European level, there is a clear association between the level of education and the social situation in which young people find themselves: “people with only basic education are almost three times more likely to live in poverty or social exclusion than those with tertiary education. In 2016, only 44.0% of young people (18–24) who had finished school below the upper secondary level were employed. And in the general population (15–64), unemployment is much more prevalent among those with only basic education (16.6%) than for the tertiary educated (5.1%)” [15]. Furthermore, it should be noted that education is essential at the personal level and at the level of development and stimulation of long-term economic growth [16]. Therefore, obtaining higher education is one of the most effective defenses against the risks of social marginalisation, poverty, and exclusion, especially at times of crisis [17].

In the current context of the COVID-19 pandemic crisis, the World Health Organization identifies young people as a priority target audience with specific concerns, experiences, and behaviors [18]. In the Official Journal of the European Union [19], it was decreed to extend the age bracket of Youth Guarantee beneficiaries as an action to ensure that this measure reaches more young people, as it is being detected that the economic recession, a consequence of the COVID-19 pandemic, will fully affect the drastic increase in unemployment among young people, thus increasing their vulnerability.

### 1.2. Young People and Social Networks during the Pandemic

The pandemic has forced people in different parts of the world to stay confined to their homes to try to reduce the number of cases of COVID-19 infection, and, consequently, internet users have turned to digital platforms and social media to keep themselves entertained [20] and at the same time to stay safe, informed and connected. According to the World Health Organization [21], young people are the most active group and interact the most daily on five digital platforms (Twitter, TikTok, WeChat, and Instagram). Meanwhile, Auxier and Anderson [22] have highlighted that most of the population uses Youtube and Facebook, while Instagram, Snapchat, and TikTok are the most common among adults under 30, with Instagram standing out as the most used.

Some research has already determined that it would be interesting to include social influencers in health campaigns to reach the maximum number of people [23]. There are already groups that use social media as the primary source of health information instead of traditional sources, such as health professionals [24,25]. Thanks to these actions carried out through social media, health information is more available, shared, and adapted [26]. Although social media is a good tool for reaching a large number of people, it should be noted that false information is tweeted more but retweeted less than science-based evidence or fact-checking tweets, while science-based evidence and fact-checking tweets capture more engagement than mere facts [27].

### 1.3. Science Education in the Social Media

Charoenwong, Kwan and Pursiainen [28] argue that people with lower levels of education are more skeptical and distrustful of science and public health recommendations in the face of COVID-19. Miller [29] and Parmet and Paul [30] point out that it is precisely this group that has ignored the recommendations made by scientists the most. This group believed that the threat of COVID-19 was a political conspiracy and is the group that is most reluctant to change their beliefs and shows their distrust of science.

Therefore, learning opportunities that help overcome mistrust of science and encourage critical thinking in science need to be encouraged and provided, offering scientific and rigorous learning opportunities [31].

Different groups, such as professionals, influencers, and accredited organisations, are already using the social networks most used by young people such as Instagram, Facebook, and Youtube to share information related mainly to health conditions and lifestyles [25,26]. 

Social media helps disseminate and present science interestingly and engagingly to people; it is worth noting that it is not necessary to have a large number of followers or to have many reactions to a post to make a difference; a single post can inspire anyone to increase their interest in science [32]. People are increasingly turning to social media as their primary source of scientific information, and therefore science education has to adapt to this new reality [33].

In fact, previous research along the same line reveals a shift in social media consumption marked by the COVID-19 pandemic, leading to an increase in citizen-driven education discussions on social media as well as Instagram often becoming the mean of education [34]. Even considering the incidence of young people’s use of social networks such as Instagram, there are studies that have tried to teach the contents of subjects in the first years of university such as histology through an Instagram account specifically for this purpose, with good results [35].

Furthermore, in the field of medicine and health, findings from similar studies regarding education through Instagram revealed that the use of certain medical and pharmaceutical hashtags could provide an opportunity to educate the public about the knowledge and skills of pharmacists [36]. 

A different study in the field of microbiology also found that social media such as Instagram is an underutilised tool for researchers and that researchers should share more content on social media, not only for the audience specifically interested in the topic, but for an increasingly global population that can benefit from that knowledge [32]. This type of research shows how this use of social media can promote scientific knowledge and reach all social media users.

## 2. Materials and Methods

This article is based on the fieldwork carried out in the ALLINTERACT project: Widening and diversifying citizen engagement in science (H2020) (ALLINTERACT is a project of the H2020 programme (SwafS-20-2018-2019: Building the SwafS knowledge base) that started in October 2020 and will end in May 2023.). ALLINTERACT is articulated around two general objectives, (1) to create new knowledge on how to transform the potential participation of citizens in science into real engagement in scientific research, and (2) to unveil new ways of involving social actors, including young citizens and groups that have traditionally been excluded from science. In achieving both goals, it aims to contribute to the achievement of two Sustainable Development Goals, “Quality Education” and “Gender Equality” [37]. 

This article applies Social Media Analytics (SMA) to analyse how social media, specifically Instagram, which is one of the most widely used by young people [22,25,26], has fostered, among other aspects, the science education of young people during the COVID-19 pandemic. 

The methodology used for this study is based on the application of (SMA), including a Communicative Content Analysis (CCA), designed for the Horizon 2020 ALLINTERACT project. For this purpose, we delve into the reading and application of the protocol established by Pulido Rodriguez, Ovseiko, Font Palomar, Kumpulainen and Ramis [38]. 

### 2.1. Research Design 

In the framework of ALLINTERACT, a protocol was created [38] stipulating and describing how to carry out the analysis of Social Media Analytics to identify and provide information on which awareness-raising actions and policies succeed in linking scientific advances with citizens. The project was analysed in March 2021, using the top-down strategy [39], the information published on gender and education in different social networks (Facebook, Twitter, Reddit and Instagram) searched for different keywords.

The selected posts were divided by social networks, an Excel document was created for each social network, the content of the post was presented in the Excel document, and the post was categorised into two sections that we present in the Table 1, the first section, and in the Table 2, the second section:

To ensure the anonymity of the posts analysed, all were first coded according to the 5 topics Table 1 and whether they were certified scientific evidence or supposed evidence. All material used for data analysis was previously collected and analyzed through ALLINTERACT project. All researchers involved in this project followed scientific and ethical procedures defined by the EU Charter of Fundamental Rights and the UNESCO Universal Declaration of Human Rights. To ensure that they cannot be identified in the results section, we do not use literal posts extracted from Instagram but report their contents. In that sense, we have ensured the compliance with the EU GDPR regulation.

### 2.2. Data Analysis

For the present research, we have collected all the work of emptying the posts on the social network Instagram related to education. The process for carrying out this search was as follows:
Search for a keyword in the browser using the symbol #Select most popular hashtags containing the keywordSearch all posts using the selected hashtags, including top and most recent posts.


The results of the search presented were as shown in Figure 1:

The criteria for selecting the hashtags were:The popularity of the hashtag (use of the hashtag and number of posts).Correspondence between the content of the post and project topics.High level of interactions among citizens (including likes and comments).

Following the established criteria, a total of 5000 posts were collected. For the present article, we selected those that were categorized in the “Scientific evidence” section as: “1. Certified Scientific Evidence” and “2. Supposed Scientific Evidence”, obtaining a total of 259 posts. From this selection, those not written in Spanish or English were discarded, as the meaning of the post’s content could not be assured, leaving a total of 233. After an in-depth reading of the 233 posts, those that responded to the article’s primary objective were selected, leaving a total of 97 posts. Of these posts, we analysed the percentage that provided information and scientific advances in various branches of knowledge and those that provided information on the COVID-19 pandemic. 

Applying the (CCA), the 97 posts were shared with the researchers through an egalitarian and intersubjective dialogue [40] and the following categories were generated for analysis: education, sociology, health, engineering, gender, science, and the United Nations Sustainable Development Goals. These categories were established after different meetings between two researchers and sharing with the rest of the researchers. Through a thorough analysis of the posts, the results shown below have been extracted, responding to the starting point of the research, which is to analyse how the social network, Instagram, has contributed to promoting, among other aspects, the scientific education of young people during the COVID19 pandemic. 

We conducted mainly qualitative analysis, but we also exploited some concrete quantitative data (Table 3). Quantitative data provides only a descriptive analysis of the percentage of messages percentages by categories. There is no information regarding possible correlations because this social media analytics under a communicative approach and applying the (CCA) presents mainly qualitative data. This content analysis is qualitative, following Pulido, Villarejo-Carballido, Redondo-Sama and Gómez [27]. We followed the seven postulates of the Communicative Methodology [41] and the principles of the egalitarian dialogue [42] during the entire process. At the same time, the communicative approach on which (CCA) is based includes the transformative and exclusionary dimensions [43]. The transformative dimension included the messages that promoted the scientific education of young people during COVID19 pandemic. Exclusionary ones were all information that stablished barriers for scientific education. We finally centred our analysis in the transformative dimension, highlighting all positive content regarding how young people exploited Instagram for scientific education purposes.

## 3. Results

From the analysis of 233 Instagram posts related to education in the categories “1. Certified Scientific Evidence” and “2. Supposed Scientific Evidence”. It has been observed that Instagram, the social network most used by young people and adolescents [21], contains information of scientific relevance. In the context of the pandemic, and mainly due to confinements, the population has turned to digital platforms and social media to stay entertained and informed, safe, and connected [17,18]. 

The results presented below are detailed in three sections. The first section presents the information and scientific advances in various branches of knowledge published on Instagram; the second section highlights how Instagram has been a social network that has provided scientific information on the pandemic. Finally, the third section identifies how Instagram has contributed to creating knowledge networks through different hashtags.

### 3.1. Information and Scientific Developments

The analysis of the posts published on Instagram shows that, of the 97 posts published on Instagram, 68 (70.10%) provide information and scientific advances related to various branches of knowledge. From these 68 posts, 32 posts (47.06%) dealt with science-related topics; 7 posts (10.19%) are linked to educational topics; 5 posts (7.35%) refer to sociology-related topics; 19 posts (27.94%) deal with health issues; 1 post (1.47%) is linked to engineering; 3 posts (4.41%) deal with gender issues, and finally, 1 post (1.47%) deals with the United Nations Sustainable Development Goals (SDGs). As presented in the Table 4.

The different categorised posts provide, in an informative way, information on additional research and or publications of a high scientific level. In this line, one of the posts published from a personal account presents scientific advances in marine biology. This specific post presents algal blooms and the causes of eutrophication, citing the article from which the information is extracted, published in the journal Nature Reviews Microbiology (a journal that occupies the first position in the area of Microbiology in the Journal Citation Reports of the ISI Web of Science). 

We also find accounts on Instagram that make playful proposals that are scientifically endorsed, as is the case of an account where topics related to technology, science, and engineering are published. In this case, it is a post where one of the most exciting books on materials science is presented. The post contains a brief review of the book and encourages its reading and scientific dissemination.

There is an essential range of posts that deal with health and provide information through a non-academic but rather informative lexicon, which aim to bring knowledge and advances made in health to all audiences. In this sense, it is worth highlighting the post where a new scientific advance is presented that, thanks to the use of machine learning and artificial intelligence, it is possible to scan the retinas of children as young as six years old to detect early autism or the risk of suffering it. In this post, the link to the article where the main results of the study are presented.

These posts exemplify the scientific information published on Instagram, which is done altruistically. Published openly, more people could access and benefit from the scientific advances and results presented in the post.

### 3.2. Pandemic Information Channel

The COVID-19 pandemic has taken a strong prominence on Instagram. Of the 97 posts selected, 29 posts (29.90%) provide information concerning the COVID pandemic. Of these 29 posts, 13 posts (44.83%) refer to COVID and science; four posts (13.79%) provide educational information about COVID; two posts (6.90%) provide information on the pandemic from a sociological perspective; eight posts (27.59%) focus on COVID and health; one post (3.45%) deals with COVID from a gender perspective, and finally one post (3.45%) deals with COVID taking into account the sustainable development goals set by the United Nations. As presented in the Table 5.

Instagram has provided rigorous information on the pandemic from different perspectives. The results of this study show some examples that stand out, such as a post by an organisation that aims to help students recognise and respond to issues that threaten their future through interactive and engaging assemblies. The post raised the results of a National Geographic article that presented a study on the impact of COVID-19 on the rise of cyberbullying. 

Posts have also been collected from accounts that have disseminated scientific articles on COVID-19, such as the case of a pediatrician account who published on his Instagram account the results of an article published in the journal Pediatrics, ranked fourth in the pediatrics section of the Journal Citation Reports. The article analysed the risk factors associated with COVID-19 in hospitalised children, specifically the study was based on a sample of 397 children who were hospitalised with severe COVID-19 infection and who had different clinical patterns. The results showed that young age is not a factor independently associated with severe COVID-19 infection and that children older than 90 days are at lower risk of severe disease progression. This study helps healthcare professionals to more easily identify the risk of severe disease progression in pediatric patients who contract COVID. Apart from presenting the main results in the Instagram post, making the link of the paper available to young people encourages its reading and subsequent dissemination, as it is also information published in a journal with a high scientific impact.

Instagram has also been used to disseminate information summarised from videos and expert panels, as is the case of an Instagram account that aims to spread the truth and debunk conspiracy. This account has posted videos accompanied by short summaries that help the reader understand the video’s subject matter and scientific lectures that may be of interest to Instagram users. For example, in this account, a post was published presenting three experts from different universities on the impact of obesity on severe complications due to COVID-19.

Therefore, Instagram has also been a channel that has provided rigorous information on COVID-19 that has helped make the population literate about the disease and its impacts at different levels.

### 3.3. Knowledge Networking

The results of this particular section focus on the exhaustive review of 5000 posts related to education. The selection of the posts was carried out taking into account the popularity of the hashtags, their similarity to ALLINTERACT project contents, and the high level of interaction among citizens, including likes and comments. These guidelines led to the analysis of 5000 posts with the hashtags #qualityeducation, #stopbullying, #learning, #scienceeducation, and #education, 1000 posts from each hashtag category. 

The hashtags are a tool for creating knowledge networks since, thanks to incorporating a hashtag in the post, it is possible for the post to enter a knowledge network, thus facilitating the search for information on specific topics. Within these hashtags, one post stands out where an organisation dedicated to promoting the diversity of careers and options that young girls have at their disposal is presented to bring information closer to this group and ensure that it impacts them. In order for this post to reach the maximum number of people interested in the topic presented, the authors incorporated a total of 29 hashtags to encourage the creation of knowledge networks and for more people interested in the topic presented in the post to find the information presented by searching for some hashtags. All of them related to the information presented in the post.

## 4. Conclusions

Through the analysis of the 5000 posts, it has been observed that Instagram, the social network most used by young people and adolescents [22], contains quality scientific information, thus providing learning spaces for different branches of knowledge when using this social network, as the scientific literature has already shown in the specific case of health conditions and lifestyles [25,26].

The results that we present in the paper have shown that on Instagram, there are posts that present research of the highest scientific level and different branches of knowledge through informative and attractive wording for the reader. This fact has already been identified in the scientific literature as key, more than the number of reactions or followers, to inspire anyone to increase their interest in science [32]. It is possible thanks to the altruism and solidarity of people who create accounts to disseminate this research and, at the same time, prepare playful proposals in their posts to capture the attention of readers.

It has also been possible to show that Instagram has been a social network on which information on the COVID-19 pandemic has been published. Although there may have been information published that did not have scientific backing, through the rigorous analysis carried out, it has been possible to identify posts that have dealt with the subject rigorously and scientifically, providing readers with up-to-date, quality knowledge, citing the sources on which they were based. This result, obtained from the exhaustive analysis carried out, was significant, as young people use Instagram as a social network that provides them with scientific information [33].

Finally, it is worth noting that Instagram has enabled the creation of knowledge networks by incorporating hashtags in the posts made by different users, thus encouraging interaction on specific topics. This fact helps people who want to obtain information on a specific topic to refine their search by using hashtags on the topic they consider.

Therefore, we can conclude that, through the innovative and incipient methodological process, not only by analysing the content of the posts, but also by the fact that we rely on the application of SMA, including a Communicative Content Analysis (CCA), designed for the Horizon 2020 ALLINTERACT project, we have analysed 5000 Instagram posts and specifically 233 that have been categorized according to: “1. Certified Scientific Evidence” and “2. Supposed Scientific Evidence”, 97 posts that respond to the main objective of the paper have been analysed and selected. 

Although it is necessary to delve deeper into the topic in question, this study shows that Instagram is a tool that during the COVID-19 pandemic has provided, in an attractive way, quality scientific information from different branches of knowledge and about the COVID-19 pandemic. This approach to science through the most widely used social network among the vulnerable group of young people has been possible thanks to the altruism of people who have decided to disseminate this information and the incorporation of hashtags in the posts, which has allowed the creation of knowledge networks.

## Figures and Tables

**Figure 1 ijerph-19-01974-f001:**
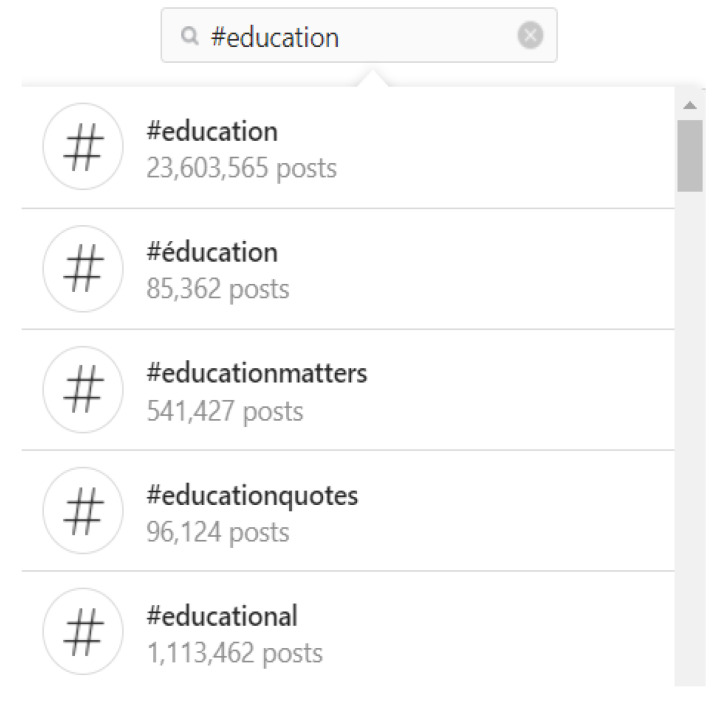
Selected hashtags.

**Table 1 ijerph-19-01974-t001:** Categories of analysis by topic.

Code	Topic	Definition
0	Not valid.	It includes those messages that are not related to project topics.
1	How citizens’ benefit from scientific research.	It includes those messages where citizens express how they implement or benefit from scientific research, although they are not aware that it is due to/do not mention scientific research.
2	Citizen awareness of the impact of scientific research.	It includes both initiatives that make citizens aware of the link between the benefits they appreciate and the research that led to them and messages where citizens express their awareness of the impact of scientific research in education and gender.
3	Awareness-raising initiatives succeeding at engaging citizens in scientific participation, including the Open Access movement.	It includes those initiatives that enhance citizen participation in science.
4	Awareness-raising actions that foster the recruitment of new talent in the sciences	It includes formal actions (e.g., campaigns, workshops, seminars, conferences, scholarships, vacancies specifically encouraging women, educational materials, interviews) and individual actions (e.g., personal experience or testimony).
5	Policies that promote awareness-raising actions and citizen engagement in science	It includes messages that mention policies, programs, or institutional measures and initiatives that promote citizen engagement in science, representation of women scientists, or the creation of spaces for the participation of vulnerable groups in science.

**Table 2 ijerph-19-01974-t002:** Categories of analysis by “Scientific evidence.

Code	Scientific Evidence	Definition
0	Not Scientific Evidence	The message does not contain scientific evidence
1	Certified Scientific Evidence	The message includes scientific evidence (articles from journals indexed in Scopus or JCR)
2	Supposed Scientific Evidence	The message says it is based on scientific evidence but no reference to the article or study.

**Table 3 ijerph-19-01974-t003:** Analysis of the posts.

5000 posts	Scientific Evidence	1. Certified Scientific Evidence	233 posts	97 posts respond to the main objective of the research
		2. Supposed Scientific Evidence		

**Table 4 ijerph-19-01974-t004:** Analysis of the posts.

97 posts respond to the main objective of the research	70.10% (68 posts) provide information and scientific progress in various branches of knowledge	47.06% (32 posts) science topics
		10.29% (7 posts) educational topics
		7.35% (5 posts) sociology topics
		27.94% (19 posts) health topics
		1.47% (1 post) engineering topics
		4.41% (3 posts) gender topics
		1.47% (1 post) United Nations Sustainable Development Goals topics

**Table 5 ijerph-19-01974-t005:** Analysis of the posts.

97 posts respond to the main objective of the research	29.90% (29 posts) provide information concerning the COVID pandemic	44.83% (13 posts) COVID and science topics
		13.79% (4 posts) educational information about COVID topics
		6.90% (2 posts) sociological pandemic perspective topics
		27.59% (8 posts) COVID and health topics
		3.45% (1 post) gender pandemic perspective topics
		1.47% (1 post) United Nations Sustainable Development Goals from a pandemic perspective

## Data Availability

The data presented in this study are openly available in Zenodo at: https://doi.org/10.5281/zenodo.4729725 (accessed on 9 September 2021).

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
