# Peer review of "An Instagram Hashtag Fostering Science Education of Vulnerable Groups during the Pandemic"

_ijerph, 2022, doi:10.3390/ijerph19041974_

Round 1

Reviewer 1 Report

1)  Is the learning objective clear and do all sections of the RLO support it?

Is Revision Required?    Yes â–ª                     No â–¡

The purpose of the study was not fully articulated.

In lines 112-119, the authors focused more on the aims of the ALLINTERACT project than on the aims of the current study.
A section of the discussion was missed
.

2)  Is the content factually correct?

Is Revision Required?    Yes â–¡                   No â–ª

Social media currently plays a very important opinion-forming role, especially among young people. Taking this aspect into account, the purpose of the article was in fact defined as: to analyze how social networks, especially Instagram, which is one of the most used by young people, promoted, among other things, scientific education of young people during the COVID19 pandemic.

3)  Is the text well written in short, clear, sentences?

Is Revision Required?    Yes â–¡                   No â–ª

For the most part, the text was comprehensible.

4)  Does the glossary cover all the terms required for a general audience?

Is Revision Required?    Yes â–¡                   No â–ª

The glossary covers the main terms necessary for a general audience.

5)  Is the structure and sequence of information helpful?

Is Revision Required?    Yes â–¡                   No â–ª

The subject of planned research is very interesting and necessary. Currently, in the era of fighting the pandemic, we can see the great impact on people of social media, both the desired messages and the undesirable messages.

6)  Are the suggestions/examples for images/animations/video appropriate?

Is Revision Required?    Yes â–ª                     No â–¡

The tables provided are not sufficient for a comprehensive study.

7)  Is sufficient interaction proposed to support active learning?

Is Revision Required?    Yes â–ª                     No â–¡

The article is a promising material, but there is a lack of more sophisticated research methods. Basing all observations only on Social Media Analytics and then expressing the dependence as a percentage raises my reservations in terms of the quality of the scientific analysis.

8)  Will the assessments measure attainment of the learning objective?

Is Revision Required?    Yes â–ª                     No â–¡

The presented material lacks, above all, advanced analytical methods such as econometric models, which could be implemented with greater care on the material presented. As I mentioned above, basing a serious scientific analysis on interest alone is a weak accomplishment.

9)  Are the keywords appropriate? Are others needed?

Is Revision Required?    Yes â–ª                     No â–¡

In general, the keywords represent the terms used in the article. However, the authors equate the concept of vulnerable groups with young people, which I do not entirely agree with.

10)         Are the suggested links OK? Are there others that you could suggest?

Is Revision Required?    Yes â–¡                   No â–ª

The literature cited in the work corresponds to the current research status in the field of the discussed issues.

11)         Have you discussed your review with the authors?

Nature of communication (eg face-to-face, e-mail etc)

No, I haven't.

Additional comments or continuations of above sections.

There is not enough information in the introduction as to whether there are publications that have already dealt with education in Instagram.

As I have already mentioned, the study deals with science education for young people during the COVID19 pandemic through Instagram. However, this does not correspond to the title of the article.

The article lacks a clear statement from the authors as to who exactly they consider to be a vulnerable group.

It is not clear to me how exactly networks are used to enhance the education of vulnerable groups. There is a lack of justification.

Lines 230-232 state that: " Making the link available to young people encourages its reading and subsequent dissemination, as it is also information published in a journal with a high scientific impact", but this statement is not substantiated. This fact is exactly what this article needs to prove.

The document is not well prepared. I propose that it be adopted after a thorough revision.

Author Response

We would like to thank the Editorial team, as well as the three reviewers, for the valuable and detailed comments on our manuscript, which have allowed us to enhance its quality. In what follows, we provide specific response to all the comments of the three reviewers, referring to the changes made and to the places in the manuscript where those changes have been made.

Thank you for the opportunity to revise and resubmit our manuscript.

Reviewer 2 Report

Dear authors,

Your research approaches an important topic nowadays when most of the people are spending a lot of time on social networks. But, in order to be published in such a prestigious journal, many improvements need to be done. First, you refer to vulnerable groups in the title, but the methodology or the results do not mention anything about this category. More, the methodology section is very vague, not mentioning anything about the target group, where the study was made, how the vulnerable group was chosen.

The introduction section do not offer sufficient information regarding similar researches conducted, taking into consideration that during the pandemic, the consumer behaviour changed drastically and it was the subject of many other researches. The results section do not comprises any figure or table in order to visualize the findings, therefore it is difficult to read.

You wrote:"Posts have also been collected from accounts that have disseminated scientific articles 226 on COVID, such as the case of a pediatrician who published on his Instagram account the 227 results of an article published in the journal "-the phrase is not scientific at all, it is vague.

lines 73-74The pandemic has forced people in different parts of the world to stay confined to their homes to try to reduce the number of cases of HIV/AIDS infection,"- i believe it is about Covid.

The discussion section is missing the results were not presented within the highlight of previous researches, and the conclusion section needs a lot of improvements since it is not clear at all the utility of the study, the impact and future directions.

The paper is poor referenced.

Good luck

Author Response

(The authors gave the same response as above.)

Reviewer 3 Report

The text of the publication is extremely interesting and up-to-date. The experiences of the COVID-19 pandemic have affected every area of daily living. The innovation of the topic should be appreciated. Applause for the authors.

Author Response

(The authors gave the same response as above.)

Round 2

Reviewer 1 Report

The paper is improved, but it still lacks more sophisticated research methods. Unfortunately I have to repeat that basing all the observations only on Social Media Analytics and then expressing the correlation in percentages makes me doubt the quality of the scientific analysis.
In addition, the authors continue to equate the notion of vulnerable groups with the notion of young people.

Author Response

The authors of the article would like to express their gratitude to the reviewers for their efforts and dedication. In relation to the link between young people and vulnerable groups, we have expanded the theoretical framework where international organizations present a concern about the situation of young people, due to the vulnerability that affects them. Concerning the analysis and treatment of the data, we have clarified in the methods section that we did not carry out any quantitative analysis. The quantitative data provided are merely descriptive to situate the reader. We have added more bibliographic sources about the communicative orientation of the analysis. We clarify that the analysis is qualitative, with a communicative orientation, which leads to communicative content analysis. We emphasize the communicative process and how communicative analysis oriented to social transformation can point out transformative results.

Reviewer 2 Report

Dear authors,

you improved your manuscript according to the indications. 

Good luck

Author Response

(The authors gave the same response as above.)
